# [Re] Reproducibility Study of Equal Improvability Fairness Notion

**Abstract**

Our research validates and expands the Equal Improvability (EI) framework, which aims to equalize acceptance rates across different groups by quantifying required improvement efforts, thereby enhancing long-term fairness. By replicating the original findings, we reaffirm the foundational claims of EI. Additionally, extended experiments are conducted to probe the efficacy of EI under varied scenarios. To enhance long-term fairness, we propose non-parametric updates and Chi-square fit to generalize the dataset, in contrast to the Gaussian distribution dataset from the original study. Our analysis shows that the EI framework struggles with adapting to the Chi-square fit and exhibits even poorer performance with non-parametric updates in long-term scenarios, indicating challenges in dynamic distribution scenarios. The update rule is modified to align more with theorem and intuition. It is proved that EI is more robust to noise compared with the other notions. The examination of varying decision fractions uncovers the conditional robustness of EI across different acceptance rates. These experiments highlight the strengths of EI in certain contexts and its limitations in others, providing a nuanced understanding of its applicability and areas for improvement in the pursuit of fairness in machine learning.

## 1 Introduction

Over the past decade, various researchers defined notions and developed classifiers for artificial intelligence to achieve fairness. For example, the fairness notion Demographic Parity (DP) is proposed to equalize the acceptance probability for different groups (Zafar et al., 2017b;a; Dwork et al., 2011; Hardt et al., 2016). While these notions focus on immediate fairness, they overlook long-term fairness. To solve this long-term unfairness issue, more fairness notions have been developed (Gupta et al., 2019; Heidari et al., 2019; von Kügelgen et al., 2022). However, these notions have various limitations, such as being vulnerable to outliers.

The paper "Equal Improvability: A New Fairness Notion Considering The Long-Term Impact" (Guldogan et al., 2023) introduces and explores the concept of Equal Improvability (EI) as a novel paradigm in the domain of group fairness. This notion enhances fairness by specifically focusing on the advancement of individuals who have been previously excluded or rejected. The research posits that EI exhibits enhanced performance and robustness compared to other fairness metrics. To verify this claim, the authors raise three approaches to tackle the EI-regularized optimization problem, which are designed to yield a model that attains EI fairness. An experimental framework is established to evaluate the efficacy of EI in comparison to other fairness notions. These experiments show the potential of EI to mitigate some of the limitations in existing fairness models. Consequently, the paper argues that the adoption of EI could ensure more enduring and equitable outcomes over time.

In this work, we investigate the reproducibility of the original paper by Guldogan et al. (Guldogan et al., 2023) by reproducing the original experiments done by the authors. Furthermore, we extend the notion into more scenarios, thereby testing its generalizability.

## 2 Scope of reproducibility

We focus on five main claims in this paper. These claims define the EI fairness and clarify its advantages by comparing with other fairness notions. EI disparity (Guldogan et al., 2023) quantifies the difference in the expected improvement after making efforts for every group, which is defined as

$$
\max_{z \in [Z]} \left| \mathbb{P}\left( \max_{\mu(\Delta \mathbf{x}_{\mathrm{I}}) < \delta} f(\mathbf{x} + \Delta \mathbf{x}) \geq 0.5 \mid f(\mathbf{x}) < 0.5, \mathbf{z} = z \right) - \mathbb{P}\left( \max_{\mu(\Delta \mathbf{x}_{\mathrm{I}}) < \delta} f(\mathbf{x} + \Delta \mathbf{x}) \geq 0.5 \mid f(\mathbf{x}) < 0.5 \right) \right| \quad (1)
$$

where sensitive attribute $z \in [Z]$ and $Z$ is the number of sensitive groups, $x$ represents the features, $\Delta x$ the efforts made by the group according to the update rule, $\mu$ a defined norm $\mathbb{R}^{d_{\mathrm{I}}} \to [0, \infty)$, and $f$ the classifier. In this paper, we adopt a binary setting for the sensitive attribute, consistent with the original paper, which categorizes individuals into two distinct groups. Additionally, the classification target is also binary, such as 'accept' or 'reject', aligning with the approach used in the original study.

The claims are listed below and studied in following sections.

- **Claim 1:** EI classifiers have good performance in balancing EI disparity and error rate, which is the proportion of all instances that were incorrectly classified by the algorithm.

- **Claim 2:** EI improves long-term fairness and accelerates mitigating long-term unfairness.

- **Claim 3:** Compared with Equal Recourse (ER, Gupta et al. (2019)), EI is more robust.

- **Claim 4:** Compared with Bounded Effort (BE, (Heidari et al., 2019)), EI is more robust.

- **Claim 5:** EI classifiers yield the lowest EI disparity when compared to Empirical Risk Minimization (ERM), ER and BE, and do not have overfitting issues.

## 3 Methodology

### 3.1 Model descriptions

In order to verify the claims mentioned above, we establish models based on the assumption and methodology of the original paper, which are listed below.

**EI fairness -** We assume each individual will improve its feature within a certain effort budget towards the direction that maximizes its score to be accepted. Therefore, EI fairness is defined as ensuring fairness in the acceptance rate across various rejected groups, considering that each individual has exerted their best effort within a predefined budget. The goal of EI fairness is to equalize the likelihood of rejected samples becoming qualified after a certain level of feature improvement across different groups, therefore promotes achieving fairness in the long-term.

Necessary features are defined to further explain the notion of EI. We sort $d$ features $\mathbf{x} \subseteq \mathbb{R}^d$ into three categories: improvable features $\mathbf{x}_{\mathrm{I}} \subseteq \mathbb{R}^{d_{\mathrm{I}}}$, manipulable features $\mathbf{x}_{\mathrm{M}} \subseteq \mathbb{R}^{d_{\mathrm{M}}}$, and immutable features $\mathbf{x}_{\mathrm{IM}} \subseteq \mathbb{R}^{d_{\mathrm{IM}}}$, where $d_{\mathrm{I}} + d_{\mathrm{M}} + d_{\mathrm{IM}} = d$ holds. It is assumed that improvable features are those can be improved and can directly affect the outcome, *e.g.*, GPA in the school's admission problem. On the contrary, manipulable features are the features can be changed, but are not directly influence the outcome, *e.g.*, marital status in the school's admission problem. Immutable features are those can not be altered. In this paper, we argue that when a sensitive attribute z is incorporated into the feature vector, it should be categorized as an immutable feature, as it holds the potential to impact fairness.

We divide samples into two categories based on their sensitive attributes. Define a norm $\mu : \mathbb{R}^{d_{\mathrm{I}}} \to [0, \infty)$. For a given constant $\delta > 0$, where a $L_n$ norm for $\mu(\mathbf{x}) = ||\mathbf{x}||_n$, a classifier $f$ is said to achieve *equal improvability with $\delta$-effort* if

$$\mathbb{P}\left(\max_{\mu(\Delta \mathbf{x}_\mathrm{I}) \leq \delta} f(\mathbf{x} + \Delta \mathbf{x}) \geq 0.5 | f(\mathbf{x}) < 0.5, \mathbf{z} = z\right) = \mathbb{P}\left(\max_{\mu(\Delta \mathbf{x}_\mathrm{I}) \leq \delta} f(\mathbf{x} + \Delta \mathbf{x}) \geq 0.5 | f(\mathbf{x}) < 0.5\right) \quad (2)$$

holds for all $z \in \mathcal{Z}$, where $\Delta(\mathbf{x}_\mathrm{I})$ is the effort made (Guldogan et al., 2023). Fig.1 shows a the geometric interpretation of EI fairness notion where the samples at the right-hand-side of the decision boundary is classified as qualified samples. $\mathbb{P}(\max_{\mu(\Delta \mathbf{x}_\mathrm{I}) \leq \delta} f(\mathbf{x} + \Delta \mathbf{x}) \geq 0.5 | f((\mathbf{x}) < 0.5, \mathbf{z} = z) = \frac{1}{3}$ holds for each group $z \in \{\text{red, blue}\}$, which satisfies EI fairness according to Equation 2.

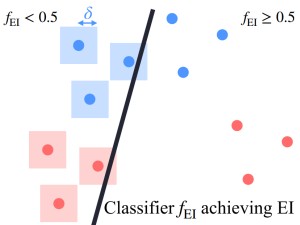

Figure 1: **EI fairness.** (Guldogan et al., 2023)

**EI classifiers -** The authors construct classifiers to achieve EI fairness by solving a fairness-regularized optimization problem. This optimization problem can be represented as

$$\max_{f \in \mathcal{F}} \left\{ \frac{(1 - \lambda)}{N} \sum_{i=1}^{N} l(\mathrm{y}_i, f(\mathrm{x}_i)) + \lambda U_\delta \right\} \quad (3)$$

where $\{(\mathrm{x}_i, \mathrm{y}_i)\}_{i=1}^N\}$ is the given dataset, $l : \{0, 1\} \times [0, 1] \to \mathbb{R}$ is the loss function, $\mathcal{F}$ is the set of classifiers we are searching over, and $\lambda$ in $[0, 1)$ is a hyperparameter that balances fairness and prediction loss . Three ways of defining the penalty term $\lambda U_\delta$ are considered, which are (a). covariance-based, (b). kernel density estimator (KDE)-based, and (c). loss-based methods. For the detail of three methods, see Appendix.A.1. Guldogan et al. (2023)

**Update data parameters -** In the original work, he authors propose that each sample can enhance its feature from $x$ to $x + \epsilon(x)$, where $\epsilon(x) = v(x; z) = \frac{1}{(\tau_t^{(z)} - x + \beta)^2} \mathbf{1}\{x < \tau_t^{(z)}\}$. This function, $\epsilon(x)$, represents the improvement of each sample's feature, with $\tau_t^{(z)}$ representing the threshold of group $z$ at time $t$ and $\beta$ being a positive constant.

In this model, the rejected samples with larger gap with the decision boundary are making less effort, which is inspired by the intuition that a rejected sample is less motivated to improve its feature if it needs to take a large amount of effort to get accepted in one scoop.

**Long-term fairness -** The EI classifier promotes long-term fairness by equalizing the feature distribution across different groups over time. In two distinct groups with varying distributions, rejected applicants in one group face a greater "improvability gap" from the decision boundary compared to the other group. This discrepancy can demotivate rejected applicants, thereby hindering their efforts to enhance relevant features over time. Consequently, the widening gap may perpetuate inequality between groups in future decisions.

To analyze fairness over time in various methodologies, long-term unfairness is quantified using the total variation distance between two groups. This measure is calculated using the Wasserstein distance (Vallender, 1974).Considering two groups, we define the long-term unfairness, denoted by $d_{TV}(\mathcal{P}^{(0)}, \mathcal{P}^{(1)})$, as

$$d_{TV}(\mathcal{P}^{(0)}, \mathcal{P}^{(1)}) = \frac{1}{2} \int_{-\infty}^{\infty} |\phi(x; \mu_t^{(0)}, \sigma_t^{(0)}) - \phi(x; \mu_t^{(1)}, \sigma_t^{(1)})| \, dx \quad (4)$$

where $\mathcal{P}^{(0)}$ and $\mathcal{P}^{(1)}$ are the probability massive function for group 0 and group 1, respectively.

## 3.2 Datasets

We used Synthetic dataset, German Credit (Hofmann, 1994) and ACSIncome (Ding et al., 2022) from the original paper. An overview of datasets is shown in Table 1.

Table 1: Dataset description

| Datasets | Samples | # Features | Sensitive Attribute |
|----------|---------|-----------|---------------------|
| Synthetic | 20,000 | 2 improvable | $y \in \{0, 1\}$ |
| German STAT. | 1,000 | 4 improvable | Age (over 30 or not) |
| ACSincome-CA | 195,665 | 1 improvable | Sex (Male or Female) |

## 3.3 Hyperparameters

The authors of the original paper provided specific hyperparameter settings for their experiments. The hyperparameters for the ACSincome-CA and German STAT datasets were not included in the code accompanying the original publication. We obtained these parameters directly from the original authors through personal communication. In our efforts to reproduce these experimental results, we chose to adhere to their established setup. For our further experiments, we maintained the same hyperparameter configuration to guarantee result comparability.

Specifically, we set true acceptance rate $\alpha = 0.2$, the maximum classification error rate $c = 0.1$, and $\beta = 0.25$ to avoid zero denominators when updating parameters under long-term settings.

Futhermore, the hyperparameters used for Synthetic Dataset is shown in Table 2

Table 2: Hyperparameter for Synthetic Dataset

| Experiment | | Chi-square | Gaussian | |
|------------|--------|------------|----------|----------|
| i | Group0 | df = 3 | $\mu = 0$ | $\sigma = 1$ |
| | Group1 | df = 4 | $\mu = 1$ | $\sigma = 0.5$ |
| ii | Group0 | df = 6 | $\mu = 0$ | $\sigma = 0.5$ |
| | Group1 | df = 7 | $\mu = 1$ | $\sigma = 1$ |
| iii | Group0 | df = 5 | $\mu = 0$ | $\sigma = 0.5$ |
| | Group1 | df = 10 | $\mu = 1$ | $\sigma = 0.5$ |
| iv | Group0 | df = 3 | $\mu = 0$ | $\sigma = 2$ |
| | Group1 | df = 8 | $\mu = 0$ | $\sigma = 1$ |

## 3.4 Experimental setup and code

Part of the code is available on the Github page of the author , while the remainder was obtained by contacting the author via email. We have re-implemented the entirety of the code of the author and have also developed new code for conducting extended experiments that substantiate our claims, all of which is accessible on our Github page.

To support Claim 1, we evaluated the error rate and EI disparities of ERM alongside three proposed EI-regularized methods applied to logistic regression (LR) across three datasets. To support Claim 2, we investigate the long-term unfairness at each round and evolution of the different feature distribution. For Claim 3, we experiment with a new synthetic dataset with more outliers. For Claim 4, we test on a synthetic dataset with varying imbalanced group negative rate. To validate Claim 5, experiments were executed using LR, multilayer perceptron (MLP), and an over-parameterized neural network as test models.

To further explore the performance of EI fairness on various scenarios, we did extensions to further challenge Claim 1 and Claim 2. We propose two methodologies for assessing the long-term fairness of EI: the Chi-

squared fit and non-parametric update methods. Firstly, we utilized a Chi-squared synthetic dataset and conducted experiments on four initial feature distribution setups. Secondly, experiments were executed on a Gaussian dataset without fitting to a Gaussian distribution, to illustrate non-parametric updating, as shown in Table 2. We also introduced a novel update rule incorporating noise to assess its impact on the robustness of the EI fairness metric. Additionally, further experiments were implemented with varying decision acceptance rates to test the generalization efficacy across different scenarios.

### 3.5 Computational requirements

Like the code of the original author, which was executed solely on a CPU, our implementation also runs on a CPU, specifically utilizing the 8-core Apple M3 chip with 16GB of unified memory. We measured the energy consumption of both CPU and RAM during model training by employing the CodeCarbon package (Courty et al., 2024).

We calculate $CO_2$ emissions in kilograms of $CO_2$-equivalents. Initially, we estimated emissions with the formula $CO_2e = CI \times PUE \times P \times t$ , accounting for Carbon Intensity ($CI$; $CO_2$ emissions per kWh), Power Usage Effectiveness ($PUE$; the ratio of total facility energy to IT energy, with 1.0 as ideal), Power ($P$; required power in kW), and time ($t$). We later simplified this to $CO_2e = CI \times E$, where $E$ is the total energy consumed in kWh, calculated as $P \times t$. For the Netherlands, we set $CI$ at 0.389 kg/kWh, based on package data, and assumed $PUE$ to be 1.0, indicating no additional energy wastage beyond IT usage.

Table 3: The carbon emissions of different models and tasks.

| MODEL/TASK | DATASET | $C0_2e$(kg) | E /kWh | Runs |
|---|---|---|---|---|
| LR & tradeoff | Synthetic | 9.80e-05 | 2.52e-04 | 5+40 |
| | German | 1.51e-05 | 3.88e-05 | 5+70 |
| | Income | 3.42e-03 | 8.78e-03 | 5+18 |
| MLP | Synthetic | 1.51e-04 | 3.88e-04 | 5 |
| | German | 1.51e-05 | 3.88e-05 | 5 |
| | Income | 2.85e-03 | 7.32e-03 | 5 |
| DNN | German | 4.30e-05 | 1.11e-04 | 5 |
| LONG-TERM | Gaussian | 1.26e-03 | 3.24e-03 | 1 |
| | non-parametric | 4.11e-04 | 1.06e-03 | 1 |
| | Chi-squared | 2.13e-04 | 5.47e-04 | 1 |

The upper section of Table 3 shows the carbon emissions of Error rate and EI disparities of ERM and three proposed EI-regularized methods on LR, MLP and deep neural network(DNN) models for one iteration. Each experiment was run five times with different seeds, and numerous preliminary experiments were conducted for hyperparameter selection. Consequently, to accurately reflect the environmental impact, the reported carbon emissions must be multiplied by five, accounting for the repeated runs and hyperparameter tuning phases. The lower section of Table 3 shows the Long-term unfairness experiments across different populations. These experiments were conducted once, and the carbon emissions listed represent the total one. The aggregate carbon emissions of experiments for our project amount to 0.260 kilograms of $CO_2$. Roughly equivalent to the carbon emissions from driving a car for 1 to 2 kilometers, or the emissions from consuming a meat-heavy meal.

## 4  Results

The results section is organized around five claims and is divided into two main parts: the replication of findings of the original paper and the new experiments that extend beyond the original work. In Section 4.1, we detail the replicated experiments from main text and appendix of the original paper. Additional experiments that expand on the original paper are presented in Section 4.2.

### 4.1  Results reproducing original paper

**Claim 1: EI Fairness -** To substantiate Claim 1, we trained the classifier using three different methods mentioned in the model description. Table 4 shows the test error rate and test EI disparity (disp.) for ERM and our three EI-regularized methods. Despite the some minor differences (highlighted in red) discovered in the original work, they are mostly within the margin of error and almost the same with those in original paper. Our experiments demonstrate that EI regularized methods successfully reduce the EI disparity without increasing the error rate too much for all three datasets.

Fig.6 shows the trade-off between the error rate and EI disparity of the EI-regularized methods. All three methods successfully find classifiers balancing EI disparity and error rate, as claimed in the original paper.

Table 4: Error rate and EI disparities of ERM and three proposed EI-regularized methods on LR. For each dataset, the lowest EI disparity (*disp.*) value is in boldface, the different value from original work is in red.

| DATASET | METRIC | ERM | METHODS COVARIANCE-BASED | KDE-BASED | LOSS-BASED |
|---|---|---|---|---|---|
| SYNTHETIC | Error Rate (%) | .221 ± .002 | .253 ± .003 | .250 ± .007 | .246 ± .002 |
| | EI Disp. (%) | .118 ± .007 | .003 ± .002 | .006 ± .004 | **.002 ± .001** |
| GERMAN STAT. | Error Rate (%) | .220 ± .009 | .262 ± .009 | .243 ± .024 | .237 ± .008 |
| | EI Disp. (%) | .041 ± .008 | .022 ± .023 | .035 ± .026 | **.016 ± .013** |
| ACSINCOME-CA | Error Rate (%) | .184 ± .000 | .200 ± .000 | .196 ± .000 | .194 ± .000 |
| | EI Disp. (%) | .031 ± .001 | .008 ± .001 | **.005 ± .001** | .006 ± .002 |

**Claim 2:  Long-term Fairness -** To verify Claim 2, we compare EI classifier with empirical ERM, DP, BE (Heidari et al., 2019), ER (Gupta et al., 2019), and individual-level fair causal recourse (ILFCR) (von Kügelgen et al., 2022). Fig. 2 shows how the long-term unfairness $d_{TV} = (\mathcal{P}_t^{(0)}, \mathcal{P}_t^{(1)})$ changes as a function of round t, cases (i)–(iv) having different initial distributions: (i).$(\mu_0^{(0)}, \sigma_0^{(0)}, \mu_0^{(1)}, \sigma_0^{(0)}) = (0, 1, 1, 0.5)$, (ii).$(\mu_0^{(0)}, \sigma_0^{(0)}, \mu_0^{(1)}, \sigma_0^{(0)}) = (0, 0.5, 1, 1)$, (iii).$(\mu_0^{(0)}, \sigma_0^{(0)}, \mu_0^{(1)}, \sigma_0^{(0)}) = (0, 2, 0, 1)$, and (iv).$(\mu_0^{(0)}, \sigma_0^{(0)}, \mu_0^{(1)}, \sigma_0^{(0)}) = (0, 0.5, 1, 0.5)$. Our reproduce result shows EI outperforms other methods in long-term unfairness. Our reproduce result in Fig.3 shows that EI brings the distribution of the two groups closer and quicker, which implies EI promotes and accelerates long-term fairness.

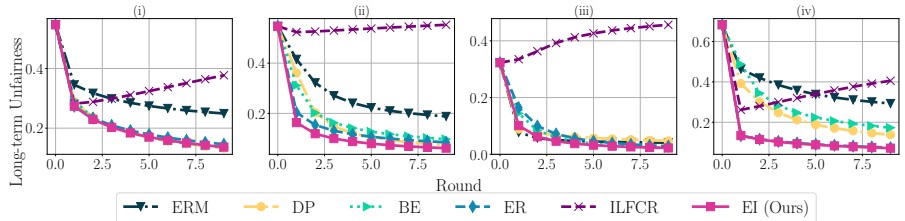

Figure 2: **Long-term unfairness** $d_{TV} = (\mathcal{P}_t^{(0)}, \mathcal{P}_t^{(1)})$ **at each round** $t$ **for various algorithms.**

**Claim 3: Outlier Robustness -** Our experimental results in Fig. 4 reveal that EI exhibits greater robustness compared to Equal Recourse (ER). Specifically, when introducing 5% outliers, the decision boundary of ER undergoes significant alterations. Adjusting the position of outliers causes the decision boundary of ER to become nearly vertical to the original one. In contrast, the decision boundary of EI is not affected, underscoring its robustness to outliers. This observation substantiates our Claim 3.

**Claim 4: Imbalanced Group -** The outcomes presented in Fig. 5 reveal that EI is more robust to imbalanced group negative rates compared with Bounded Effort (BE). The decision boundary of BE exhibits substantial rotation in response to varying negative rates within the dataset. Furthermore, as we further adjust the negative rates, BE continues to display variability. In contrast, the consistent behavior of decision

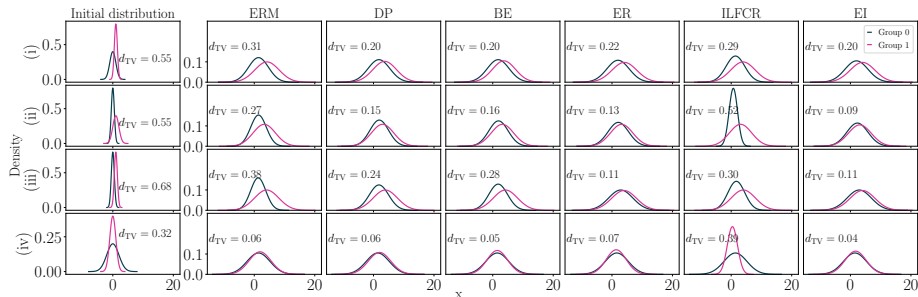

Figure 3: **Evolution of the feature distribution, when we apply each algorithm for t = 3 rounds**

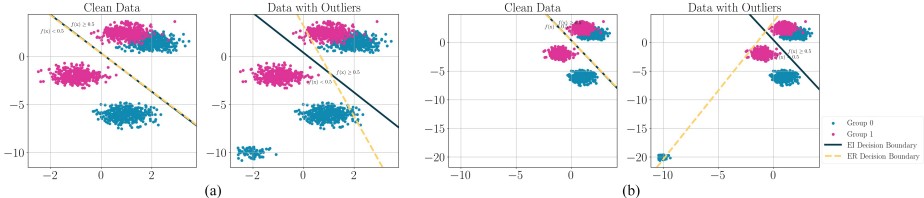

Figure 4: **Visualizations of the EI and ER decision boundaries without and with the presence of outliers.** (a). Reproduce result with 5% outliers added. (b). The position of outliers are adjusted.

boundaries of EI underscores its robustness to imbalanced negative rates. These findings support our Claim 4.

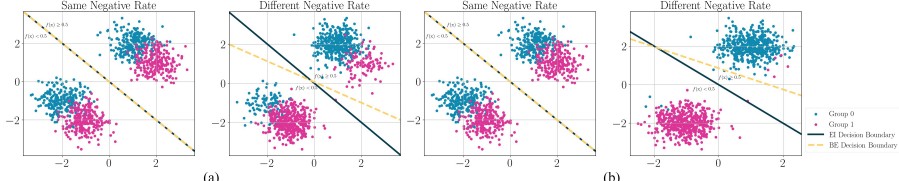

Figure 5: **Visualizations of the EI and ER decision boundaries with the same negative rates and different negative rates.** (a). The reproduce result of original paper, where y|z = 0 ∼ Bern(0.8), and y | z = 1 ∼ Bern(0.2). (b). y|z = 0 ∼ Bern(0.99), and y | z = 1 ∼ Bern(0.01).

**Claim 5: Robustness in MLP and over-parameterized networks -** The performance of the proposed models on both MLP and over-parameterized neural networks, as detailed in Tables 6 and 7, supports Claim 5. Despite minor discrepancies from figures of the original paper, it is evident that the EI fairness classifiers achieve high EI fairness and low error rates.

This consistent performance across various neural network architectures also affirms the robustness of the EI fairness approach against overfitting.

## 4.2   Results beyond original paper

**Feature Distribution -** The long-term feature distribution is tested on a synthetic dataset with Gaussian distribution originally. In the original paper, the feature distributions of two groups tend to be the same in the long run with EI fairness. This is only tested in Gaussian distribution. And after every round, the distribution is fitted into Gaussian distribution with two parameters. Possibly, the feature adhere to different distributions other than Gaussian. To further challenge Claim 1, we investigate the performance of EI fairness feature distribution on non-Gaussian distributions.

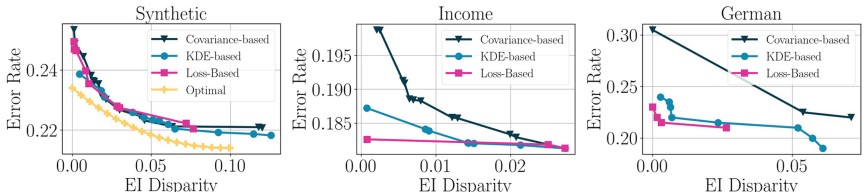

Figure 6: **Tradeoff between EI disparity and error rate.**

Table 5: Comparison of error rate and EI disparities of ERM baseline and proposed methods on the synthetic, German Statlog Credit and ACSIncome-CA datasets on Multi-Layer Perceptron (MLP).

| DATASET | METRIC | ERM | Covariance-Based | KDE-Based | Loss-Based |
|---|---|---|---|---|---|
| SYNTHETIC | Error Rate (%) | .215 ± .003 | .242 ± .006 | .227 ± .007 | .229 ± .012 |
| | EI Disp. (%) | .141 ± .036 | .004 ± .003 | .011 ± .006 | .019 ± .009 |
| GERMAN STAT. | Error Rate (%) | .221 ± .010 | .300 ± .010 | .235 ± .020 | .238 ± .035 |
| | EI Disp. (%) | .059 ± .046 | .000 ± .000 | .019 ± .007 | .013 ± .019 |
| ACSINCOME-CA | Error Rate (%) | .182 ± .002 | .203 ± .002 | .183 ± .002 | .187 ± .002 |
| | EI Disp. (%) | .038 ± .003 | .010 ± .011 | .008 ± .004 | .003 ± .003 |

To simulate different scenarios, we propose two ways: keep the feature distribution untouched, or fit it into a different distribution other than Gaussian.

In the initial test case, we utilized a non-parametric method that does not presuppose a specific distribution for updated sample features; instead, it directly compares the distribution curves between groups. For example, the updated parameters $(\mu_{t+1}^{(z)}, \sigma_{t+1}^{(z)})$ are not assumed to be the input for the distribution of the subsequent round. The starting distributions are identical to those used in the Claim 2 experiment. Methods relying on the ILFCR, which are dependent on Gaussian distribution parameters, were excluded. Figure 7 illustrates that data points within the acceptance region remain unchanged. In contrast, points that are proximate to but do not transgress the boundary show more pronounced shifts compared to those further away, which manifest minimal movement. Upon examining long-term unfairness, the observed patterns did not align with the original patterns depicted in Fig. 2. We attribute this divergence to the overlapping of data points that traverse the boundary line, posing a challenge in precisely evaluating the strengths and weaknesses of various methods using our unfairness metric (refer to Eq. 4).

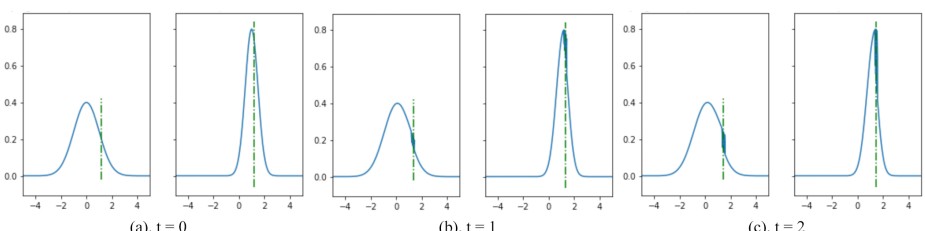

Figure 7: **The long-term progression of a non-parametric distribution from round $t = 0$ to round $t = 2$.** Following the formula of authors, $\epsilon(x) = \nu(x; z) = \frac{1}{(\tau_t(z) - x + \beta)^2} \mathbf{1}\{x < \tau_t(z)\}$.

In the second test case, we assume a chi-square distribution to simulate the distribution of features in different groups, as shown in Fig. 9. The updating rule for parameters is as follows:

$$k_{t+1}^{(z)} = \int_{-\infty}^{\infty} (x + \boldsymbol{u}(x; z))\phi(x; k_t^{(z)})dx \tag{5}$$

where $\phi(\cdot; k)$ is the pdf of $\chi^2(k)$.

Table 6: Error rate and EI disparities for ERM baseline and proposed methods, for an over-parameterized neural network on German Statlog Credit dataset. Performances on train/test dataset are presented.

| DATASET | METRIC | ERM | Covariance-Based | KDE-Based | Loss-Based |
|---|---|---|---|---|---|
| GERMAN STAT. | Train Err. ($\downarrow$) | $.218 \pm .004$ | $.233 \pm .003$ | $.225 \pm .009$ | $.232 \pm .011$ |
| | Test Err. ($\downarrow$) | $.218 \pm .010$ | $.218 \pm .010$ | $.221 \pm .010$ | $.230 \pm .009$ |
| | Train EI Disp. ($\downarrow$) | $.022 \pm .017$ | $.018 \pm .011$ | $.018 \pm .009$ | $.015 \pm .013$ |
| | Test EI Disp. ($\downarrow$) | $.060 \pm .032$ | $.049 \pm .024$ | $.057 \pm .028$ | $.047 \pm .025$ |

Table 7: Comparison of error rate and EI disparities of ERM, ER, and BE baseline and proposed methods on the synthetic dataset.

| METRIC | ERM | ER (Gupta et al., 2019) | BE (loss-based) | Covariance Based | KDE-Based | Loss-Based |
|---|---|---|---|---|---|---|
| Error Rate (%) | $.221 \pm .002$ | $.235 \pm .009$ | $.252 \pm .006$ | $.253 \pm .003$ | $.250 \pm .007$ | $.246 \pm .002$ |
| EI Disp. (%) | $.118 \pm .007$ | $.036 \pm .018$ | $.006 \pm .004$ | $.003 \pm .002$ | $.006 \pm .004$ | $\mathbf{.002 \pm .001}$ |

In alignment with the approach of original author, we analyzed the binary classification task within two separate groups under dynamic conditions. The long-term unfairness for all classifiers decreases, except when the parameter $k$ is small for both groups, which leads to an increment in unfairness for certain metrics, EI included. The results indicate that for synthetic datasets that follow a Chi-square distribution, the performance of EI is not superior to other metrics.

From the above experiment, we found that the unfairness measurement method (Eq. 4) might not effectively compare the strengths and weaknesses of different methods in non-parametric fitting scenarios. Additionally, the EI measurement exhibits poor performance on chi-square distributed data. This finding suggests that EI may perform better on Gaussian data than on other distributed data.

**Update feature -** In the original work, the authors used $\epsilon(x) = v(x; z) = \frac{1}{(\tau_t^{(z)} - x + \beta)^2} \mathbf{1}\{x < \tau_t^{(z)}\}$ to model the improvement of each sample on its feature. This corresponds with the Expectancy-Value Theory in psychology (Wigfield, 1994), assuming that the amount of realized effort is inversely proportional to the required amount of effort, or expectancy, to improve the outcome. However, the effort is also proportional to the reward or value it will receive by making such effort, which is a factor missing in the original formula. Additionally, since each sensitive group has different mean and variance, reflecting distinct characteristics, we assume that each sensitive group also has a different reward for making an effort. Considering individual variability, we introduce an additional noise term that depends on the sensitive group.

To further investigate Claim 2, we enhance the update formula as:

$$\epsilon(x) = v(x; z) = (\underbrace{\frac{1}{(\tau_t^{(z)} - x + \beta_e)^{\gamma_e}}}_{\text{Expectancy}} \underbrace{(\mu_t^{(z)} - x + \beta_v)^{\gamma_v}}_{\text{Value}} + \underbrace{\gamma_n \sigma_t^{(z)}}_{\text{Noise}}) \mathbf{1}\{x < \tau_t^{(z)}\} \tag{6}$$

Where $\gamma_e, \gamma_v$ represent the power of the expectancy term and value term respectively, and $\gamma_n$ control the scale of the noise term. $\beta_e, \beta_v$ are used to bound the corresponding terms.

We conducted a similar task as in the previous section. With the inclusion of additional noise terms, the long-term unfairness curve exhibited increased variability across all groups, as illustrated in Figure 10 and detailed in Appendix. A.2. We found that EI fairness could mitigate long-term unfairness in all test cases. Although EI does not consistently outperform alternative methods, it generally demonstrates superior performance on average due to its robustness against noise.

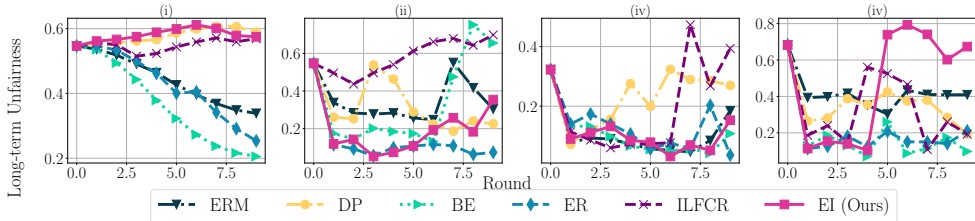

Figure 8: **Long-term unfairness** $d_{TV} = (\mathcal{P}_t^{(0)}, \mathcal{P}_t^{(1)})$ **at each round** $t$ **for various algorithms with non-parametric distribution.** See experimental settings in Table 2.

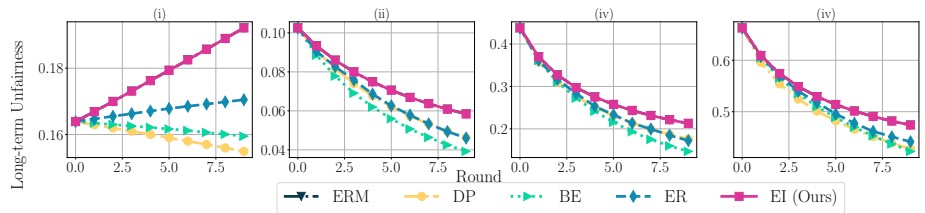

Figure 9: **Long-term unfairness** $d_{TV} = (\mathcal{P}_t^{(0)}, \mathcal{P}_t^{(1)})$ **at each round** $t$ **for various algorithms under the assumption that the initial data (and the updated data after each round) follow the Chi-squared distribution with degree of freedom** $k$**.** See experimental settings in Table 2.

**Decision fraction -** As shown in Fig. 3, the feature distributions of the two groups become identical in the long term and thus improve long-term fairness. To further investigate the generalizability of this effect, we varied the decision fraction in the model applied to a synthetic dataset. The decision fraction is defined as the acceptance rate within the model, which is relevant to various scenarios, such as differing acceptance rates across universities. The term 'distance' refers to the Wasserstein distance, utilized here to quantify the disparity between two distributions. We conducted experiments on a synthetic dataset characterized by normal distributions $\mathcal{N}(0,1)$ and $\mathcal{N}(1,0.5)$.

The evolution of feature distributions across different models is illustrated in Figure 11. By comparing the performance of different models, it can be concluded that EI fairness is robust to different decision fraction setting, though it does not outperform the ER model all the time.

In conclusion, the analysis of EI fairness across different decision fraction settings demonstrates that EI is a versatile and robust fairness framework, capable of adapting to a wide range of application scenarios. Although EI does not consistently surpass the ER model in every case, its general resilience in varying conditions highlights its potential as a valuable tool in promoting fairness in algorithmic decision-making processes.

## 5    Discussion

This paper undertakes a comprehensive analysis by reproducing and extending the work "Equal Improvability: A New Fairness Notion Considering the Long-Term Impact." This study reproduces the findings of the original paper, confirming the significance of Equal Improvability (EI) in mitigating bias. Additionally, this study conduct supplementary experiments to delve deeper into the efficacy of the EI framework. While the reproduced results align closely with the original findings, the new experiments shed light on potential weaknesses in the EI method, particularly when applied to synthetic data with altered distribution assumptions. By rigorously testing the proposed definition of long-term fairness and exploring scenarios beyond Gaussian distributions, this study provides valuable insights into the robustness and limitations of the EI framework.

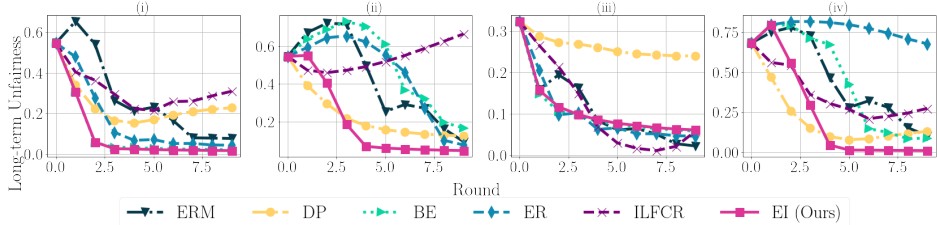

Figure 10: **Long-term unfairness** $d_{TV} = (\mathcal{P}_t^{(0)}, \mathcal{P}_t^{(1)})$ **with new update rule, noise** $\gamma_n = 1$, $\gamma_v = 1$

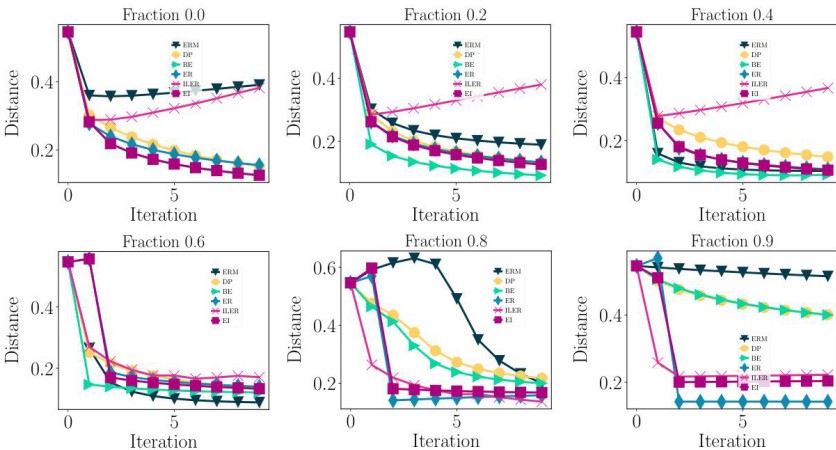

Figure 11: Comparison of different models on long-term fairness with varying decision fraction

## 5.1   what was easy

The code of the original paper is well organized and easy to run. The emphasis on graphical representation in various script sections enhances the interpretability of the employed methodologies. This aspect is especially pronounced in the creation process of the synthetic dataset, where the steps are delineated in a manner that is both transparent and visually intuitive. Consequently, this approach significantly aids in the reproducibility of the dataset, thereby facilitating the consistent duplication of the experimental results.

## 5.2   what was difficult

We observed that varying versions of library packages noticeably influence the final results. Therefore, adhering to the original environment settings as closely as possible is crucial for replicating the intended outcomes. However, our attempts to replicate the original environment faced challenges due to conflicting dependencies and compatibility issues. In some cases, we needed to rebuild the environment from scratch to ensure optimal performance and functionality.

It is hard to interpret and understand why generalizing the long-term fairness method to other datasets with different distributions shows better results for Gaussian distributions. Figure 13 shows the variation of the long-term Gaussian fitting distribution with iterations (rounds $t$). At $t = 0$, we observe the initial distribution, and at $t = 1$, we see the distribution after the next update. We see the Gaussian distribution has changed at $t = 1$, with both the median and standard deviation increasing, and the distribution of the right-side group changing more significantly. These calculations are derived from the formulas mentioned in the text of the author $\mu_{t+1}^{(z)} = \int_{-\infty}^{\infty} (x + \nu(x; z))\phi(x; \mu_t^{(z)}, \sigma_t^{(z)})dx$ and $\sigma_{t+1}^{(z)} = \sqrt{\int_{-\infty}^{\infty} (x + \nu(x; z) - \mu_{t+1}^{(z)})^2 \phi(x; \mu_t^{(z)}, \sigma_t^{(z)})dx}$, which seems mathematically sound. However, original authors did not elaborate on why this is reasonable. For instance, we do not understand why the distribution must still follow a Gaussian pattern after the update, why points that have already entered the acceptance region continue to advance, or why points far from the

boundary line and not crossing it move towards negative values. To a certain extent, these phenomena do not align with the intuition mentioned in the original paper. When we extended this part of the experiment, we were not entirely certain that the updated population should follow a specific distribution. Besides, we tried to conduct non-parametric experiments as we want to align with this intuition. But we struggled with how to deal with points that have just crossed the boundary line and are clustering near it. And it is challenging to interpret the unusual population depicted in Figure 7 and the long-term evolution of unfairness shown in Figure 8. The unusual results obtained with the Chi-squared method (fitting the updated features to a new Chi-squared distribution parameterized with $k$) shown in Figure 9 are also related to this difficulty.

## 5.3 Communication with original authors

For acquiring the hyperparameters relevant to the other two datasets, we reached out to the author of the study. They responded promptly and graciously, providing us with the necessary codes.

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

## A   Appendix

### A.1   Methods for Finding Classifiers Achieving EI Fairness

The three methodologies proposed in the study are designed to identify classifiers that adhere to Equal Improvability (EI) fairness. This is achieved by calculating EI fairness through three distinct approaches, each offering a unique perspective on how to integrate and measure EI principles within classifier systems.

**Covariance-based EI penalty** This penalty computes fairness by measuring the covariance between a sensitive attribute and the score of a classifier, ensuring demographic parity. EI unfairness is quantified by the covariance between the sensitive attribute and the maximum score improvement of rejected samples within a given effort budget. The penalty is squared to also penalize negative correlation, offering a measure of EI unfairness. The EI unfairness is then approximated by the square of the empirical covariance.

**KDE-based EI penalty** The paper proposes a KDE-based EI penalty approach, inspired by Cho et al. (2020), for estimating the fairness of a classifier. This method involves approximating the probability density function of a score of classifier using a kernel density estimator. The estimator is then integrated into the unfairness penalty formula. The maximum score improvement for each feature within a predefined effort budget is computed, and the density of the improved scores for unqualified samples is approximated using the kernel density function. The probability terms in the definition of EI fairness are estimated through the densities, and the EI penalty is computed as the summation of the absolute differences of these probabilities across all sensitive groups.

**Loss-based EI penalty** This penalty calculates the absolute difference of group-specific losses, aligning with the EI fairness concept. It quantifies the distance that rejected samples of a group are from being accepted post feature improvement within a set budget. The overall EI loss is the aggregate of these individual losses across all groups, and the EI penalty is defined as the summation of absolute differences between the group losses and the overall loss.

### A.2   Experiment: Update feature

In this section, we experiment with different amount of noises. Specifically, we test hyperparameter $\gamma_n = 0$, 0.1, 0.5 and 1, see 12 and 10. We found as more noises are added, the curves become more fluctuated. EI does not consistently surpass alternative methods, for example, when noise=0, in (ii), the performance of DP is better than EI. Nevertheless, on average, EI demonstrates superior performance due to its robustness with noises, since the performance of EI does not change much with different level of noise.

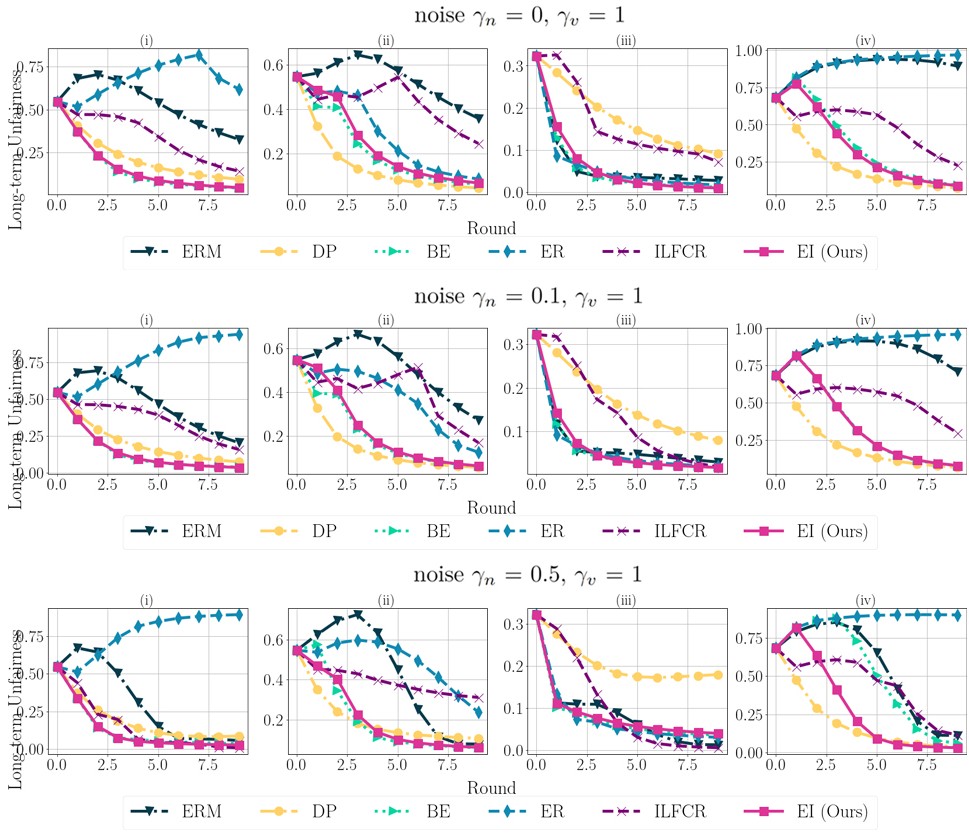

Figure 12: Long-term unfairness $d_{TV} = (\mathcal{P}_t^{(0)}, \mathcal{P}_t^{(1)})$ with new update rule

## A.3   Long-term Gaussian fitting distribution variation with iterations

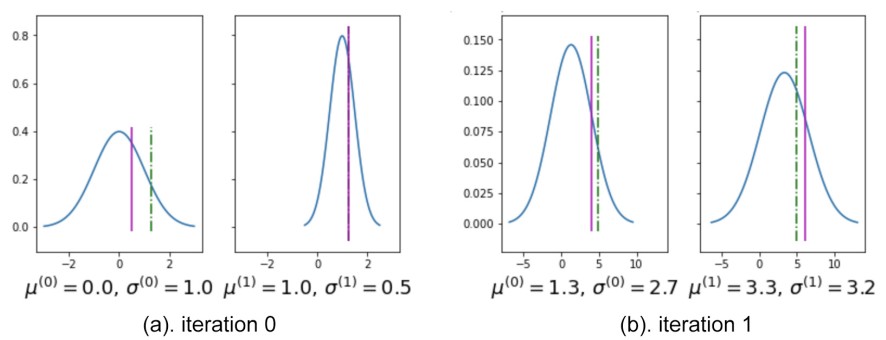

Figure 13: This is the graph from the code of the author and settings showing the long-term Gaussian fitting distribution variation with iterations (rounds $t$). The green dashed line represents the boundary for empirical risk minimization (ERM), which does not have fairness constraints. The purple solid line represents the boundary calculated by EI. At $t = 0$, we see the initial distribution, and at $t = 1$, the moment after the next update.

### A.4 Long-term unfairness extra experiment on COMPAS dataset

We conducted an additional experiment on long-term fairness using the real-world Recidivism Racial dataset (COMPAS) (Ofer, 2016), and selected race as the sensitive attribute with group 0 as Caucasian and group 1 as African-American. The score is designated initially on a scale from 1 to 10 (integer), as the improved feature, where lower scores signify a reduced risk of reoffending. Unlike the previous case, the goal here is to achieve lower scores. Figure 14 shows the initial distribution between the two groups, highlighting existing fairness between two groups. To update individual scores, we used the equation $\epsilon(x) = \nu(x;z) = -\frac{1}{(\tau_t(z) - x + \beta)^2}\mathbf{1}\{x > \tau_t(z)\}$, avoiding fitting the updated scores with specific parameters or distributions due to the sparsity of discrete scores. As illustrated in Figure 15, EI performed comparably to ERM and outperformed other methods. We plotted our analysis to the first five rounds as further updates made scores turn into very small negative values, possibly due to the discontinuous data and sparsity. Given the limited project time, we think this exploration is a good preliminary trial on real datasets, and need further more experiments.

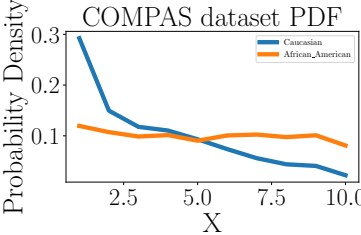

Figure 14: The population of real-world Recidivism Racial dataset (COMPAS) (Ofer, 2016) for two groups: group 0 as Caucasian and group 1 as African-American. The score is designated initially on a scale from 1 to 10 (integer), as the improved feature, where lower scores signify a reduced risk of reoffending.

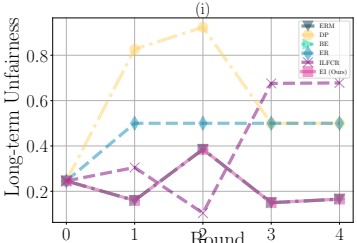

Figure 15: Long-term unfairness at the first five rounds for various algorithms for COMPAS dataset. EI performed comparably to ERM and outperformed other methods.

