# OpenReview forum: "[Re] Reproducibility Study of Equal Improvability Fairness Notion"
_TMLR — Rejected by TMLR_

### Review · Reviewer_86HD · 2024-04-01

**Summary Of Contributions:**

The paper presents a reproducibility study on the ICLR 2023 paper: "Equal Improvability: A new fairness notion considering the long term impact". The authors first introduce the same concepts introduced in the original paper, then they reproduce the results of the paper and finally they further test the proposed definition of long-term fairness by dropping some of the assumptions made in the experimental analysis of the original paper (e.g., they investigated the performance of EI fairness on non-Gaussian distributions).

**Audience:**

Yes

**Broader Impact Concerns:**

No concerns

**Claims And Evidence:**

No

**Requested Changes:**

Please make the introduction to the concepts much more precise, see the comments above.

**Strengths And Weaknesses:**

Strengths:
- Conducting reproducibility studies is of outmost importance for the community.
- Going beyond the experimental analysis given in the original paper is valuable as it proves the applicability of the proposed paper.

Weaknesses:
- The main weakness of the paper is the lack of rigour in which the method is presented at the very beginning. The authors do not mention the difference between *immutable, manipulable and improvable" features, which is actually central for the definition of the framework.
- In definition (1) $x$ should actually be $\textbf{x}_I$, thus representing an improvable feature. However, the authors write that $x$ is a sensitive feature.
- In definition (2) then $\textbf{x}_I$ appears and it is referred as improvable features. Shouldn't it be the same as the $x$ used in definition (1)?
- The definition given of EI disparity in the paper seems different, what is the reason for this difference?
- In definition (1) $\mu$ should be defined. What does it represent?

Minor problems:
- there is some formatting issue in euwaltion (2)
- f is not slanted above equation (2)
- in equation (1), $Z$ needs to be defined, then $z$ just represents an element in that set
- PUE appears sometimes slanted and sometimes not slanted, and same for $E$

---

### Review · Reviewer_QvMi · 2024-04-10

**Summary Of Contributions:**

This is a reproducibility study of Guldogan, et al. (2023), which proposed the notion of Equal Improvability (EI). Informally speaking, EI aims to capture the long-term impact of bias mitigation by equalizing the efforts needed by each demographic subgroup to alter the prediction of the classifier $f(x)$ when changing their features $x\to x+\Delta_x$ (e.g. acquiring a college degree). The paper uses the source-code released by the original authors and replicated their findings, with substantial agreement overall. In addition, they conducted new experiments on synthetic data (e.g. by using a mixture of Gaussians or a Chi-square distribution instead of a single Gaussian distribution for each group). The authors claim that the main findings no long hold in the new settings.

**Audience:**

No

**Claims And Evidence:**

No

**Requested Changes:**

- Please see my comments above about the presentation of the paper, specifically:
  * Fixing typos.
  * Providing a full description of the assumed setup from the beginning of Section 2 (e.g. cardinality of the sensitive attribute and label space).
  * Explain how the parameter are updated with each round of the algorithm. This is important because a substantial portion of the paper uses those intermediate values without defining them in the paper.
  * Describing in details the experimental setup used in the new experiments.
  * Key equations are incorrect, and some symbols are never defined.
  * The values highlighted in red (to indicate they were different from the original paper) are mostly within the margin of error so they are not really different.

**Strengths And Weaknesses:**

*Strengths*

1- EI is an interesting new notion of fairness. The paper confirms that the main findings of the original paper can be replicated using the original paper's released source code.

2- The authors go slightly beyond the original setup in the synthetic data experiments. Assuming that the experimental setup is correct, the findings about the sensitivity to the Gaussian assumption would be important to keep in mind.

*Weaknesses*

1- The paper is poorly written. In many places, there are typos, incorrect equations, or wrong statements, and the exact experimental setup is unclear. For example:
  * The abstract alone contains two typos: "algin" --> "align" and "experiments and highlight" --> "experiments highlight".
  * The authors do not describe the full setup. For instance, it seems that this is restricted to binary classification, although this is not mentioned. Also, the definition in (1) doesn't make an assumption on the cardinality of the sensitive attribute, but the definition of "long-term unfairness" assumes it is binary. A full description of the assumed setup should be included at the beginning.
  * The equation for EI disparity is incorrect as far as I can tell. This is not equivalent to Definition 2.1 in the original paper. Also, the equation for long-term unfairness is incorrect; it should be w.r.t. to the features after each round of update (subscripted by $t$). Otherwise, Equation 4 is not impacted by the choice of the classifier.
  * There are undefined symbols. For example, $\mu$ in Equation (1) is undefined so it's unclear what the maximization is taken over. Also, the authors condition on $z=z$, which is mathematically meaningless. It should be $\textbf{z}=z$ or $Z=z$ (where the first is the random variable and the second is the instance).
  * The authors mention that $z$ represents "different groups" while $x$ "represents the value of sensitive attribute". This is wrong. $x$ is the original features.
  * The authors state in Page 2 that "sensitive features are those can not be altered." This seems to be incorrect to me, and contradicts the experiments reported in the original paper. For example, ACSINCOME-CA has one improvable feature (education) but also has only one sensitive attribute (perceived sex). Where is this mentioned in the original paper?
  * The discussion in some places is unnecessarily lengthy. For example, in Page 4, the authors say that they calculate the carbon footprint using $CI\times  PUE \times P \times t$ and then use the next three lines to basically say that they assume $PUE=1$ (they introduce a symbol $E$ and redefine carbon footprint as $C\times E$, where $E=P\times t$ and $PUE$ is set to unity ...)
  * The paper does not talk about the different rounds and how the parameters are updated. The first mention of this is in Page 5 without any definitions. This makes the subsequent discussions about the convergence of the distributions of instances difficult to parse, without referring to the original paper.
  * Please check if the symbols in Page 7 in $\mu^{z}t+1$ and $\sigma^zt+1$ are correct.
  * In Page 8, the authors say in one sentence that "EI fairness could mitigate long-term unfairness in all test cases" and in the next sentence say the opposite, that "EI does not consistently surpass alternative methods". And then in the following sentence, they say that "EI demonstrates superior performance"!!

2- In some places, the claims about having "different value from original work" are incorrect. For example, in Table 2, the numbers highlighted in red are almost identical (within the margin or error) to what were reported in the original paper.

3- My biggest concern is regarding the new experiments. The authors replicate the findings of the original authors, but claim that those findings do not hold when using, for example, a mixture of Gaussians. However, the authors in the original paper did use real-world datasets, and I doubt that those datasets would adhere to the Gaussianity assumption. In the present paper, it is unclear what exactly was done by the authors in the new experiments because they do not explain it in sufficient detail to see if any modifications would be needed.

---

### Review · Reviewer_9pNt · 2024-05-06

**Summary Of Contributions:**

This paper reproduces the recent paper “Equal Improvability: A New Fairness Notion Considering the Long-Term Impact”, and also proposes a few additional experiments to better understand the efficacy of the Equal Improvability (EI) framework.  The authors were able to reproduce the findings of the original work, which reaffirms the claims made in the original paper as to the value of EI.  Some of the additional experiments highlight potential weaknesses of the EI method.

**Audience:**

Yes

**Claims And Evidence:**

No

**Requested Changes:**

Overall I think this paper needs some substantial reworking in order for me to recommend acceptance.  I think some rewriting and additional background on EI would be helpful to introduce the method.  Some additional discussion on summarizing the findings of the reproducibility study and on the new experiments would also help.   One of the most confusing aspects was on the distinction between the notion of long-term fairness and how that is assessed and evaluated, vs the earlier sets of experiments around looking at error rates and EI gap for a single frozen model. The subsection on the “Decision fraction” was also hard to follow and I’m not sure what the takeaways from that are.

Additional new experiments, maybe using new datasets, would also improve the quality of this work by showing that the EI generalizes to other settings beyond just the original 3 datasets shown in the original paper.

**Strengths And Weaknesses:**

The paper appears to have done a thorough job at reproducing the results in the original paper, and made some minor tweaks to some of the original experimental setups that confirm the original findings. In some new experiments, the authors show that in simulations of the long-term fairness setting where feature distributions change over time that EI sometimes performs poorly.

Overall, the main weakness of the paper is that it is somewhat hard to follow and I found myself constantly needing to refer to the original paper for clarity.  The exposition introducing EI was especially hard to follow.  In addition, the discussion is somewhat brief and didn’t do much for me at summarizing what the most important takeaways are from this work in reproducing and adding onto the original EI paper.

---

### Decision · Action_Editor_sU2V · 2024-06-23

**Recommendation:** Reject

**Comment:**

For reasons stated above, the clarity of claims and evidence presented need improvement.

**Audience:**

Some individuals in TMLR's audience would be interested a reproducibility study of Guldogan, et al. (2023).

**Claims And Evidence:**

A major concern shared by reviewers is the clarity of presentation of this work.  Details are sparse, and readers need to hunt for potential answers in the original study.  Reviewers note that some definitions are incorrect and terms are misused, clouding any conclusions drawn.  This lack of clarity undermines the link between claims made and evidence presented.